# Significance of Direct Confirmation of Growth Hormone Insensitivity for the Diagnosis of Primary IGF-I Deficiency

**DOI:** 10.3390/jcm9010240

**Published:** 2020-01-16

**Authors:** Joanna Smyczyńska, Urszula Smyczyńska, Maciej Hilczer, Renata Stawerska, Andrzej Lewiński

**Affiliations:** 1Department of Endocrinology and Metabolic Diseases, Polish Mother’s Memorial Hospital-Research Institute, 93-338 Lodz, Poland; maciej.hilczer@neostrada.pl (M.H.); renata.stawerska@icloud.com (R.S.);; 2Department of Biostatistics and Translational Medicine, Medical University of Lodz, 92-215 Lodz, Poland; urszula.smyczynska@umed.lodz.pl; 3Department of Pediatric Endocrinology, Medical University of Lodz, 93-338 Lodz, Poland; 4Department of Endocrinology and Metabolic Diseases, Medical University of Lodz, 93-338 Lodz, Poland

**Keywords:** growth hormone (GH), GH insensitivity, insulin-like growth factor-I, IGF-I deficiency, short stature

## Abstract

Primary insulin-like growth factor-I (IGF-I) deficiency is a synonym of growth hormone (GH) insensitivity (GHI), however the necessity of direct confirmation of GH resistance by IGF-I generation test (IGF-GT) is discussed. GHI may disturb intrauterine growth, nevertheless short children born small for gestational age (SGA) are treated with GH. We tested the hypothesis that children with appropriate birth size (AGA), height standard deviation score (SDS) <−3.0, GH peak in stimulation tests (stimGH) ≥10.0 µg/L, IGF-I <2.5 centile, and excluded GHI may benefit during GH therapy. The analysis comprised 21 AGA children compared with 6 SGA and 20 GH-deficient ones, with height SDS and IGF-I as in the studied group. All patients were treated with GH up to final height (FH). Height velocity, IGF-I, and IGF binding protein-3 (IGFBP-3) concentrations before and during first year of treatment were assessed. Effectiveness of therapy was better in GHD than in IGF-I deficiency (IGFD), with no significant difference between SGA and AGA groups. All but two AGA children responded well to GH. Pretreatment IGF-I and increase of height velocity (HV) during therapy but not the result of IGF-GT correlated with FH. As most AGA children with apparent severe IGFD benefit during GH therapy, direct confirmation of GHI seems necessary to diagnose true primary IGFD in them.

## 1. Introduction

In 1966, Laron et al. [1] published the first report concerning consanguineous children with extremely short stature despite high growth hormone (GH) secretion. Although the described pathology had to be determined genetically, in that time it was unclear if it was related to abnormal GH molecule or to resistance of target organs to GH [2]. During the next 20 years it was explained that this condition was caused by defects of GH receptor, and thus it was classified as GH insensitivity (GHI) [3]. As the main laboratory findings in this disease include insulin-like growth factor-I (IGF-I) deficiency (IGFD) and normal GH levels, in current classifications it is labelled as primary IGFD, which is considered to be synonymous of GHI, while secondary IGFD is equivalent to GH deficiency (GHD) [4]. Moreover, it turned out that disorders of GH–IGF-I axis should be considered as a continuum from severe GHD to complete GHI, including a wide range of less severe impairments of GH secretion or GH sensitivity, with different phenotypic and biochemical features, related to a variety of genetic defects [5,6]. Up to now, more than 90 different mutations of GH receptor have been described [7]. Except for the classic form of GH receptor resistance, currently known as Laron syndrome, other molecular defects leading to reduced GH sensitivity have been described, including ones in the intracellular GH signaling pathway (STAT5B, STAT3, IKBKB, IL2RG, PIK3R1), synthesis of insulin-like growth factors (IGFs) and disorders of IGF-I – insulin-like binding protein-3 – acid-labile subunit (IGFALS) complex [8,9,10]. As intrauterine growth is stimulated mainly by IGF-I and insulin, with no important role of fetal pituitary GH for human growth before birth, only the latter of these defects (related to IGF-I synthesis and to formation of IGFALS complexes) may disturb not only postnatal but also intrauterine growth [8]. Despite the fact that in Laron’s first studies defects of GH molecule were excluded [3], the possibility of bioinactive GH secretion (Kowarski syndrome) has been included in “ESPE classification of pediatric endocrine diagnoses”, however as a relatively rare condition [4].

According to previous recommendations of Blum et al. [11], the diagnosis of severe primary IGFD in a patient with normal GH peak in stimulation tests (stimGH) required performing IGF-I generation test (IGF-GT) with short-term recombinant human GH (rhGH) administration and demonstrating IGF-I increase by less than 15.0 ng/mL during this test, and an increase of IGF binding protein-3 (IGFBP-3) by less than 400.0 ng/mL. In recent years, other definitions of primary IGFD have been introduced, based on low height standard deviation score (hSDS), low IGF-I levels (below the established cut-off values), and normal stimGH, with no direct emphasis on documenting GHI [12,13]. The significance of IGF-GT has been questioned by some researchers [5,6,13], who have pointed at its limited usefulness, especially with respect to diagnosing less severe (partial) GHI. Nevertheless, other authors still consider IGF-GT as required to differentiate primary IGFD from other diseases, like idiopathic short stature (ISS) or decreased GH bioactivity [14,15].

It is also known that the majority of IGF-I in circulation is bound in IGFALS complexes, while only free IGF-I is biologically active. So, IGF-I/IGFBP-3 ratio is considered as a marker of IGF-I bioavailability [16]. Previous studies have shown the increase of IGF-I/IGFBP-3 molar ratio in children with short stature, both in IGF-GT and during the initial period of rhGH therapy [16,17].

In Europe, children with normal stimGH can be treated with rhGH only if they are born small for gestational age (SGA) with no need to exclude GHI, while ones with appropriate birth size (AGA) are either diagnosed with GHI and treated with recombinant IGF-I (rIGF-I) or diagnosed with idiopathic short stature (ISS) and untreated. According to US guidelines, only children with GHI of proven, genetically determined cause should be qualified directly to rIGF-I therapy, while in those with unexplained IGFD, a trial of rhGH administration is recommended. Moreover, US children with ISS may be qualified to rhGH therapy [18].

Having regard to the problems discussed above, we decided to test the hypothesis that AGA children with short stature, severe IGFD, and excluded GHI may benefit during rhGH therapy despite normal stimGH.

## 2. Materials and Methods

### 2.1. Patients’ Inclusion Criteria

Taking into account the ongoing debate on the diagnostic criteria of severe primary IGFD, a retrospective analysis was performed in GH-treated children who fulfilled currently proposed criteria of severe primary IGFD according to Savage et al. [6], however did not fulfil the classic criteria of GHI according to Blum et al. [11]. Such a group is to some extent unique, as all the patients were treated with rhGH before introduction of rhGH therapy for SGA children and rIGF-I therapy for ones with severe primary IGFD. Thus, rhGH in doses used for GHD was the only growth-promoting therapeutic possibility in them.

Inclusion criteria comprised the following:very short stature, defined as hSDS below −3.0, according to Polish reference charts [19];normal GH peak (over or equal to 10.0 μg/L) in at least one of the stimulation tests (stimGH): with clonidine (0.15 mg/m^2^, orally, GH concentrations measured every 30 min during 120 min) and with glucagon (30 μg/kg intramuscularly, not exceeding 1000 μg, GH concentrations measured in 0, 90, 120, 150, and 180 min;severe IGFD, defined as IGF-I concentration below 2.5 centile for age and sex, according to reference data provided by Elmlinger et al. [20];significant IGF-I increase (at least doubling the initial IGF-I concentration, leading to its normalization) in IGF-GT performed during 7 days, with a daily rhGH dose of 0.033 mg/kg;persistence of severe IGFD, confirmed by decreased IGF-I concentration (below 2.5 centile) in second measurement after at least 6 months;excluded chronic conditions that may influence both GH and IGF-I secretion, that is, other hormonal deficits, malnutrition, neoplastic processes, diagnosed or suspected skeletal dysplasiae, glucocorticosteroid administration, and other therapies that may disturb the function of somatotropic axis;normal thyroid function;normal female karyotype (46,XX) in girls;excluded neoplastic processes in hypothalamic–pituitary region (based on magnetic resonance imaging);known gestational age (GA) and birth size;known body mass at therapy onset;known Tanner’s pubertal stage;bone age (BA), assessed according to Greulich-Pyle standards [21];known target height (TH), calculated from parental heights;known height velocity (HV) before treatment;rhGH therapy with standard doses used for children with GHD, up to the attainment of final height (FH);known HV and IGF-I concentration in first year of treatment;known IGFBP-3 concentrations in all the assessed time points.

According to birth size, the patients fulfilling the inclusion criteria were classified into two groups: AGA-IGFD and SGA-IGFD. The comparative group included children with GHD, fulfilling the same inclusion criteria as the studied group, except for decreased stimGH (below 10.0 µg/L) and with no IGF-GT performed.

The following age-dependent variables were expressed as standard deviation scores (SDS): height SDS according to Polish reference data [19], birth weight and length SDS according to Niklasson et al. [22], IGF-I SDS according to Elmlinger et al. [20]. Bone age delay with respect to chronological age (CA) was calculated as BA/CA ratio. For assessment of nutritional status of patients, pretreatment body mass index (BMI) was calculated for each child and compared with reference data [19] adjusted for height age of patients in order to avoid misclassification of patients whose weight is appropriate for height but not for age.

The study was approved by the Committee of Bioethics in Polish Mother’s Memorial Hospital—Research Institute.

### 2.2. Indices of rhGH Therapy Effectiveness

According to current guidelines [23], the attained FH is considered as the main clinical endpoint of rhGH treatment, while HV during first year of treatment as a surrogate endpoint; additionally IGF-I concentration during rhGH therapy is considered as a biomarker of GH action, while IGF-I/IGFBP-3 ratio as a marker of IGF-I bioavailability. In the present study, the following indices of rhGH therapy effectiveness have been evaluated:the attained FH SDS,corrected FH SDS (corr FH SDS), calculated as the difference between FH SDS and TH SDS,increase of FH SDS with respect to pretreatment height SDS (ΔhSDS),increase of IGF-I SDS, of IGF-I/IGFBP-3 mola ratio, and of HV in first year of treatment.

### 2.3. Laboratory Assays

The concentrations of GH were measured by the two-site chemiluminescent enzyme immunometric assay (hGH IMMULITE, DPC) for the quantitative measurement of human GH, calibrated to WHO IRP 98/574 standard, with the analytical sensitivity up to 0.01 μg/L, the calibration range up to 40.0 μg/L, the intra-assay coefficient of variation (CV) of 5.3–6.5%, and the inter-assay CV of 5.5–6.2%. Serum total IGF-I and IGFBP-3 concentrations were assessed by a solid-phase, enzyme-labelled chemiluminescent immunometric assay (IMMULITE, DPC). The assay for IGF-I was calibrated to WHO NIBSC 1st IRR 87/518, with analytical sensitivity 20.0 μg/L, the calibration range up to 1600.0 μg/L, the within-run CV 3.1–4.3%, and the total CV 5.8–8.4%. The assay for IGFBP-3 was calibrated to WHO NIBSC Reagent 93/560, with analytical sensitivity 0.1 μg/mL, the calibration range up to 16.0 μg/mL, the within-run CV 4.1–4.8%, and the total CV 5.2–7.3%. For calculation of IGF-I/IGFBP-3 molar ratio, the following molecular masses were used: 7.5 kDa for IGF-I and 42.0 kDa for IGFBP-3.

### 2.4. Statistical Analysis

First, baseline characteristics of three patients’ groups (AGA-IGFD, SGA-IGFD, and GHD) were compared. Fisher exact test was used for assessment of differences in proportion of males and females in particular groups, while Kruskal–Wallis test was selected for continuous comparisons of variables, because pretreatment characteristics usually did not follow normal distribution in at least one group (verified by Shapiro–Wilk test). If significant differences were detected by Kruskal–Wallis test, it was followed by Dunn post hoc test in order to identify which pairs of groups differed.

Differences in treatment effectiveness indices between groups were also assessed with Kruskal–Wallis test. Two consecutive measurements in the same group (e.g., pretreatment height SDS vs. FH SDS in the same patients) were compared by Wilcoxon test; *t*-test for repeated measurements was not chosen since normality assumption was often violated. When more than two repeated assessments took place (IGF-I: pretreatment, at the end of IGF-GT and in first year of therapy), Friedmann test was used, and when it yielded significant result, it was followed by Nemenyi post hoc test. All correlations between variables were assessed by nonparametric Spearmann’s method. Analysis was performed in Python and R. Always *p*-value below 0.05 was considered statistically significant.

## 3. Results

### 3.1. Characteristics of Patients

In a database containing more than 1000 patients treated with rhGH in our center during the last 20 years, 27 children fulfilling the inclusion criteria to the studied group were found, among them there were 21 AGA and 6 SGA ones (all SGA with birth weight below −2.0 SD for gestational age and normal birth length) as well as 20 children with GHD, fulfilling the inclusion criteria to the comparative group.

There were no significant differences between baseline characteristics of these three groups, except for those resulting from different inclusion criteria for particular groups, that is, significantly lower birth weight in SGA-IGFD group and significantly lower stimGH in GHD group. Taking into account the possible influence of nutritional status on IGF-I levels, it seems important to state that all children had BMI over third centile for height age. Detailed pretreatment characteristics of all groups are presented in Table 1.

### 3.2. Effectiveness of rhGH Therapy

In the first year of treatment, height velocity (HV) was significantly better in GHD than in SGA-IGFD (*p* = 0.009), while in AGA-IGFD it was insignificantly better than in SGA-IGFD but slightly worse than in GHD. There was no significant difference between the groups in HV increase with respect to pretreatment values (ΔHV), as well as in IGF-I SDS and IGF-I/IGFBP-3 molar ratio in the first year of therapy. In all groups, the increase of final height (FH) SDS with respect to pretreatment (baseline) hSDS (Figure 1) was significant (*p* = 0.0001 for AGA-IGFD, *p* = 0.03 for SGA-IGFD, and *p* = 0.0001 for GHD in Wilcoxon test). Despite the fact that the pubertal stage of children before treatment was appropriate for their height age and delayed bone age rather than to chronological age, in all of them puberty started spontaneously, and at the attainment of FH, the patients reached at least Tanner’s stage IV.

On the contrary, all the intergroup differences concerning the indices related to FH were insignificant. Interestingly, despite shorter therapy duration, all these indices were slightly better in AGA-IGFD than in SGA-IGFD. In all the Groups, mean FH exceeded −2.0 SD, which is considered within normal range. Moreover, in AGA-IGFD and in GHD, the same threshold was exceeded even by lower bound of 95% confidence interval, from which we can conclude that treatment actually results in normalization of patients’ height in adulthood. Detailed data on all treatment-related variables are presented in Table 2.

### 3.3. IGF-GT Results and Effectiveness of rhGH Therapy in AGA-IGFD Group

In the AGA-IGFD Group, the increase of IGF-I during IGF-GT was observed in every single patient and ranged from 51.3 to 514.0 ng/mL. Comparison of IGF-I SDS at three timepoints—before IGF-GT, at the end of IGF-GT, and during first year of rhGH treatment—showed significant changes (*p* < 0.0001 in Friedmann test) in IGF-I SDS after administration of rhGH (Figure 2). During IGF-I GT, the increase was on average equal to 2.72 SD (*p* = 0.001 in Nemenyi post hoc test), while during first year of treatment, IGF-I increased by 3.50 SD with respect to pretreatment values (*p* = 0.001). Further increase of IGF-I was observed in the majority of patients between the end of IGF-GT and the first year of therapy, but this difference was not statistically significant. Pretreatment IGF-I/IGHBP-3 molar ratio in this Group was very low and increased significantly (from 0.13 to 0.42) in the first year of treatment (*p* = 0.0001).

Changes in HV for particular patients during the first year of therapy, being the first directly measurable outcome of treatment (surrogate endpoint), are illustrated in Figure 3. An increase in HV equaled on average 4.8 cm/year and was significant in the whole group with *p* < 0.0001 in Wilcoxon test. However, two patients (marked in red in Figure 3) appeared distinctly different than the rest of the group with HV increase not exceeding 1 cm/year. The pattern of changes in height from pretreatment value to final height was very similar to changes in HV, what is visible in Figure 4. Final height SDS also strongly and significantly correlated with HV in the first year of treatment in this group of patients (Spearman’s correlation coefficient r = 0.60, *p* = 0.004), which suggests its predictive value. Pretreatment IGF-I correlated negatively with FH SDS (r = −0.52, *p* = 0.016), while correlations between FH SDS and either maximum of IGF-SDS in IGF-GT or IGF-I SDS increase during this test were insignificant.

Two girls from AGA-IGFD Group and one from SGA-IGFD Group had FH SDS below −3.0, however in SGA one—whose age was relatively advanced at therapy onset (13.6 years)—pretreatment hSDS was very low (−4.46) and the increase of hSDS was, in fact, quite good (1.07 during 2.2 years of treatment). Clearly, two girls from AGA-IGFD group (marked in red in Figure 4) were nonresponders, whose hSDS almost did not change during treatment (see Table 3 with detailed characteristics of those patients). Moreover, they were exactly the same children whose HV did not improve much during the first year of treatment, which suggests that such patients could be identified at initial period of rhGH administration. Interestingly, in those two cases, lack of response to treatment in terms of improved height was not reflected by IGF-I values that were relatively high comparing with the rest of group and followed typical scheme of changes after rhGH administration (see red lines in Figure 2).

## 4. Discussion

The results obtained in the present study indicate that only auxological parameters (pretreatment hSDS, HV in first year of rhGH therapy) and IGF-I SDS before treatment, but not the results of IGF-GT, were the predictors of the attained FH SDS. Considering the reasons for this situation, the controversies concerning IGF-GT in the literature should be kept in mind. This test was introduced in order to differentiate patients with GH resistance from GH-sensitive ones, however, in case of so-called partial GHI it presented to have some limitations [11,24,25,26,27]. In several studies, IGF-GT was recommended as an useful tool to predict the growth response to rhGH therapy in GH-sensitive children [17,28,29], however, the poor reproducibility of this test has also been raised [30,31]. On the other hand, the study suggesting that the result of IGF-GT may be a marker of GH secretory status has also been published [32]. It seems that, despite the limitations of IGF-GT, the diagnosis of severe primary IGFD should be confirmed by the poor IGF-I increase in response to rhGH administration.

In the studied group, hSDS increase during treatment was better in children with more severe deficit of height and more severe IGFD, however with no significant relationship with either stimGH or the result of IGF-GT. Children identified as poor responders to rhGH therapy had good IGF-I response in IGF-GT but no improvement of HV in the first year of therapy. It is impossible to draw general conclusions from two cases, but careful monitoring of growth response to treatment may be of particular importance in identifying nonresponders rather than relying on normalization of IGF-I levels only. Similar observations concerning the poor response to rhGH administration in a proportion of children with good IGF-I increase in first year of treatment have been reported in a recent study of Straetmans et al. [33]. It seems that such patients should be candidates to molecular tests, however, in case of our girls, relatively low values of TH SDS should be taken into account.

In Poland, only children with GHI confirmed by IGF-GT have been qualified to rIGF-I therapy. Very recently, Petriczko et al. [34] have published the data on FH of the first 27 patients with severe primary IGFD who completed treatment and attained FH. The average increase of hSDS in them was 1.45. It is very difficult to speculate what could be the effectiveness of rIGF-I therapy in our patients from AGA-IGFD group if currently proposed diagnostic criteria of primary IGFD [6,13] were applied, but their average increase of hSDS during rhGH treatment was 2.08. It should be stressed that despite the fact that all children in AGA-IGFD group fulfilled the diagnostic criteria of severe primary IGFD proposed by Savage et al. [6], almost all of them presented to be good responders to rhGH therapy, that speaks against the diagnosis of GHI in them. It seems that for our AGA-IGFD patients, the classic diagnostic criteria of primary IGFD, provided by Blum et al. [11], could be more appropriate.

In our study, there was also no significant difference in rhGH therapy effectiveness between SGA-IGFD and AGA-IGFD children with normal GH secretion, with even slightly better response to treatment in AGA-IGFD than in SGA-IGFD ones. The conclusions, however, must be cautious due to a relatively small number of SGA-IGFD patients. It has previously been reported that effectiveness of rhGH administration in children born SGA is independent from their GH secretory status [35]. Moreover, Derraik et al. [36] have documented reduced GH sensitivity associated with a progressive decrease in birth weight, which seems to explain the possibility of no worse response to rhGH therapy in AGA children than in SGA ones.

The results of present study, obtained in a specific group of children with severe deficit of height and severe IGFD, correspond to that obtained in our neural model of rhGH therapy effectiveness for children with short stature and wide spectrum of GH and IGF-I secretion, in which neither GH peak after pharmacological stimulation nor birth weight presented to be predictors of the attained FH [37].

In Poland, for several years children born SGA have received a chance of rhGH treatment in spite of normal stimGH, while ones born AGA are usually not subjected to rhGH therapy. Treatment with rIGF-I is much more expensive, more onerous, and burdened with more serious side effects than rhGH therapy. There is also compelling evidence that GH has some IGF-I-independent effects on growth plate chondrogenesis and longitudinal bone growth [38,39]. Exogenous rIGF-I administration leads to suppression of GH secretion that precludes direct GH effects on tissues. A case report of a Polish GH-deficient patient who improved growth during rhGH therapy despite very limited increase of serum IGF-I, explained by IGF-I-independent effects of rhGH on growth plates, was published in 2017 by Flader et al. [40]. Thus, we suggest to treat GH-sensitive children with rhGH but not with rIGF-I. Our position is in line with current recommendations of Grimberg at al. [18], who have proposed starting from rIGF-I replacement only in patients unresponsive to rhGH due to known hormone signaling defects, while in unexplained cases of IGFD, a trial of rhGH therapy has been suggested. An important argument against starting treatment in children with unknown molecular cause of IGFD from rIGF-I is the observation that patients with so-called partial GHI may benefit during rhGH therapy similarly to those with mild GHD [41].

Another difficulty is an arbitrarily defined cut-off level for normal and subnormal stimGH, independent from the applied pharmacological agents but—in different studies—established on the level of 10 µg/L or 7 µg/L [12,42,43,44,45,46]. Additionally, different cut-off values for both height SDS and IGF-I level have been used by different authors to define primary IGFD [6,12,14], and the significance of direct confirmation of GHI is still a matter of controversy [6,14,18,47]. In this situation, it is possible to diagnose the same patient either with primary IGFD (i.e., GHI) or with secondary IGFD (i.e., GHD). The clinical implications of the established diagnosis are related to the possibility and the type of treatment—with either rhGH or rIGF-I. As our results confirm effectiveness of rhGH treatment in the vast majority of short children with severe IGFD and good IGF-I response in IGF-GT, we are convinced that there is no reason to deprive them from such treatment just because they are born AGA and have GH peak after pharmacological stimulation over the assumed cut-off level.

In our opinion, the patients with severe IGFD should not have the diagnosis of ISS assigned and remain untreated, as deficit of IGF-I is a known cause of growth retardation, regardless of whether it is primary or not. According to “ESPE classification of paediatric endocrine diagnoses” [4], the diagnosis of ISS may be established in a patient with “no evidence of endocrine deficiencies”. Until the cause of IGFD is documented, one of possibilities is decreased bioactivity of endogenous GH, however it is also important to remember that the diagnostic criteria of GHD have evolved over time [44,48] and that in population there exists a continuum of GH–IGF-I axis defects causing short stature [5]. The need to create a classification that covers a wide spectrum of patients with IGFD was raised by Cohen in 2006 [49]. The possibility of successful rhGH therapy in children with IGFD and normal stimGH has been reported in several studies, in which the authors have diagnosed their patients as either having ISS [14,50] or secreting bioinactive GH [15]. It seems that IGFD in children with normal stimGH and excluded GHI may be also related to inappropriate GH secretion for the attainment of normal concentration of IGF-I. This situation is to some extent similar to secondary deficiencies of peripheral glands, in which secretion of tropic hormones is within reference range but it appears to be too low for normal secretion of peripheral hormones. The differences are related to the pulsatile manner of GH secretion and the necessity of its assessment in stimulation tests, as well as to the fact that IGF-I is not secreted by an endocrine gland. We suggest to qualify such patients as ones with secondary IGF-I deficiency or to create a separate diagnostic category: non-primary IGF-I deficiency. The specific criteria of this diagnosis should be defined after conducting further studies on this issue, while our results may be regarded as preliminary.

## 5. Conclusions

Children with severe deficit of height, normal GH peak after pharmacological stimulation, and severe IGF-I deficiency, in whom GHI is excluded by IGF-I increase in generation test, may benefit during rhGH therapy, regardless of appropriate or small birth size for gestational age. Assessment of GH sensitivity in such patients should be taken into account as currently, SGA children are qualified to rhGH therapy (with no recommendation to exclude GHI), while AGA ones may be diagnosed in different countries with either primary IGFD (if IGF-GT is not recommended) or ISS (if IGF-GT is obligatory and its result excludes GHI). In our opinion, they have a form of non-primary IGFD that should be included in the classification of GH–IGF-I axis defects leading to short stature.

## Figures and Tables

**Figure 1 jcm-09-00240-f001:**
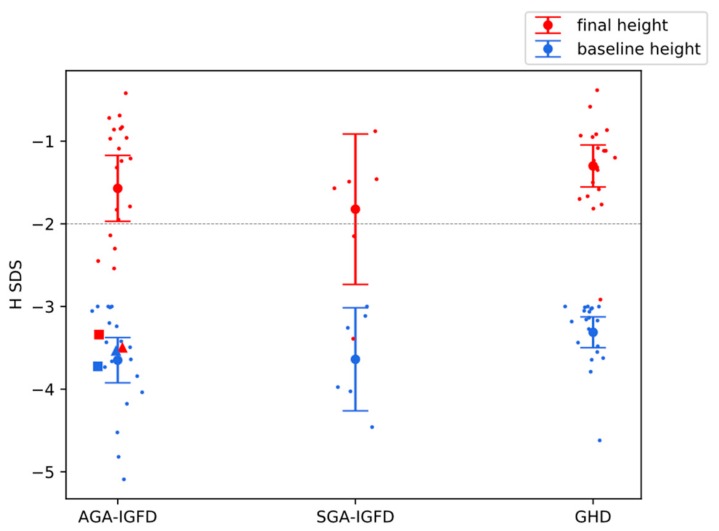
Patients’ height before (blue) and after (red) rhGH (recombinant human growth hormone) treatment, shown as mean with 95% confidence interval, with dots representing single patients. Triangle and square in AGA-IGFD group mark nonresponder 1 and 2, respectively (Table 3).

**Figure 2 jcm-09-00240-f002:**
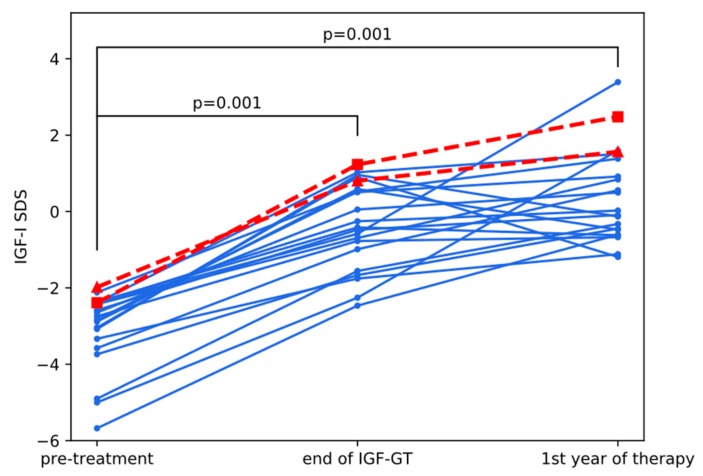
Changes in IGF-I SDS during IGF-I GT and first year of rhGH treatment in AGA-IGFD children. Red lines with triangle and square markers indicate nonresponders 1 and 2, respectively (Table 3). All significant *p*-values from Nemenyi post hoc test are shown.

**Figure 3 jcm-09-00240-f003:**
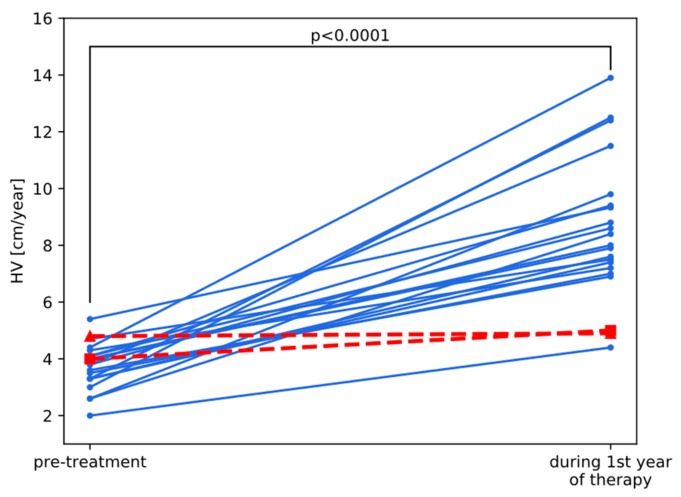
Comparison of pretreatment and final height in AGA-IGFD children. Red lines with triangle and square markers indicate nonresponders 1 and 2, respectively (Table 3). *P*-value from Wilcoxon test.

**Figure 4 jcm-09-00240-f004:**
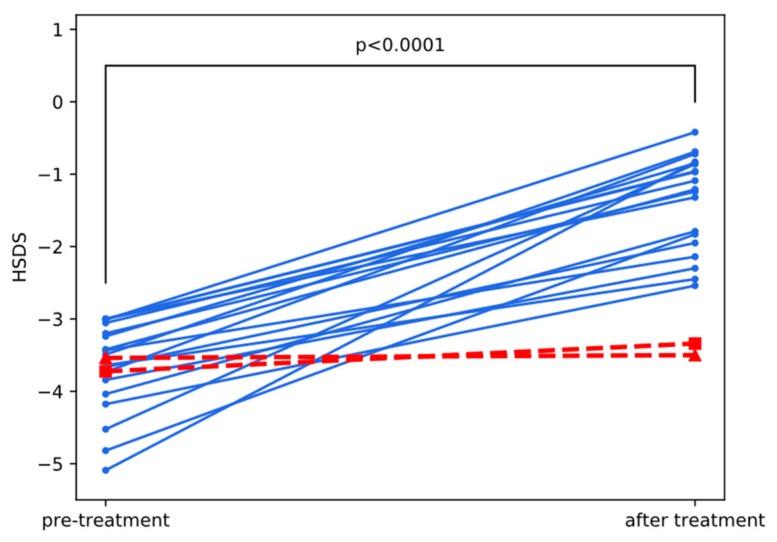
Comparison of pretreatment and final height in AGA-IGFD children. Red lines with triangle and square markers indicate nonresponders 1 and 2, respectively (Table 3). *P*-value from Wilcoxon test.

**Table 1 jcm-09-00240-t001:** Pretreatment Characteristics of Particular Groups of Patients.

	AGA-IGFD	SGA-IGFD	GHD	*p*
Number of patients (boys/girls)	21(17/4)	6 (5/1)	20 (14/6)	0.72 ^a^
Tanner’s pubertal stage				
I	17 (15/2)	6 (5/1)	13 (8/5)	
II	3 (2/1)	0	5 (4/1)	
III	3 (2/1)	0	2 (2/0)	
Age (years)	13.0 ± 1.9	12.4 ± 2.5	13.0 ± 0.8	0.96
Gestational age (GA) (weeks)	39.6 ± 1.1	38.3 ± 3.1	39.0 ± 1.1	0.28
Birth weight SDS	−0.60 ± 0.78	−3.08 ± 0.32	−0.88 ± 1.18	<0.001 ^b^
Pretreatment hSDS	−3.65 ± 0.60	−3.64 ± 0.59	−3.31 ± 0.40	0.16
TH SDS	−1.14 ± 0.80	−1.06 ± 1.00	−1.13 ± 0.48	0.82
Corr hSDS	−2.51 ± 0.85	−2.58 ± 0.96	−2.18 ± 0.54	0.27
Pretreatment HV (cm/year)	3.7 ± 0.8	3.4 ± 1.0	3.6 ± 1.2	0.43
Pretreatment BMI (kg/m^2^)	18.2 ± 3.7	15.0 ± 2.4	17.6 ± 3.0	0.12
BA/CA ratio	0.80 ± 0.08	0.74 ± 0.11	0.78 ± 0.09	0.31
StimGH (µg/L)	18.9 ± 8.4	14.7 ± 8.6	7.6 ± 2.0	<0.0001 ^c^
IGF-I SDS before IGF-GT	−3.07 ± 1.00	−2.60 ± 0.39	−3.20 ± 1.01	0.57
IGF-I SDS at the end of IGF-GT	−0.35 ± 1.13	−0.19 ± 0.62	N/A	0.44
IGF-I SDS increase in IGF-GT	2.72 ± 0.76	2.41 ± 0.67	N/A	0.18
Baseline IGF-I/IGFBP-3 molar ratio	0.13 ± 0.05	0.17 ± 0.07	0.16 ± 0.04	0.27

^a^ Fisher exact test. ^b^ Significant differences between GHD and SGA-IGFD (*p* = 0.0014), AGA-IGFD and SGA-IGFD (*p* = 0.0002). ^c^ Significant differences between GHD and AGA-IGFD (*p* < 0.0001), GHD and SGA-IGFD (*p* = 0.0036). N/A—not applicable, IGF-GT was performed only in the patients with normal GH secretion. AGA-IGFD: patients with insulin-like growth factor-I deficiency born appropriate for gestational age, SGA-IGFD: IGFD patients born small for gestational age, GHD: growth hormone deficiency, SDS: standard deviation score, hSDS: height SDS, TH SDS: target height SDS, Corr hSDS: corrected hSDS, HV: height velocity, BMI: body mass index, BA: bone age, CA: calendar age, StimGH: GH peak in stimulation tests, IGF-I: insulin-like growth factor-I, IGF-GT: IGF-I generation test, IGFBP-3: insulin-like growth factor-binding protein 3.

**Table 2 jcm-09-00240-t002:** Data Concerning rhGH Therapy Effectiveness in Particular Groups of Patients.

	AGA-IGFD	SGA-IGFD	GHD	*p*
HV in first year of treatment (cm/year)	8.5 ± 2.5	7.2 ± 2.2	9.2 ± 1.4	0.03 ^a^
ΔHV (cm/year)	5.0 ± 3.2	3.7 ± 2.7	5.7 ± 2.2	0.13
IGF-I SDS in first year of treatment	0.43 ± 1.21	0.34 ± 0.51	0.32 ± 0.74	0.89
IGF-I/IGFBP-3 molar ratio in first year of treatment	0.42 ± 0.08	0.42 ± 0.15	0.50 ± 0.21	0.78
Therapy duration (years)	4.1 ± 1.3	5.2 ± 3.1	4.8 ± 0.8	0.15
FH SDS	−1.57 ± 0.88	−1.82 ± 0.87	−1.30 ± 0.54	0.40
Corr FH SDS	−0.43 ± 1.01	−0.76 ± 0.81	−0.17 ± 0.62	0.26
ΔhSDS	2.08 ± 0.98	1.82 ± 0.53	2.01 ± 0.52	0.69

^a^ Significant difference between GHD and SGA-IGFD (*p* = 0.009). ΔHV: pretreatment height velocity, FH SDS – final height SDS, ΔhSDS: increase in height SDS during GH treatment.

**Table 3 jcm-09-00240-t003:** Detailed Data of Nonresponders to rhGH Therapy.

		Nonresponder 1 (Triangle in Figure 1, Figure 2 and Figure 3)	Nonresponder 2 (Square in Figure 1, Figure 2 and Figure 3)
Pre-treatment data	Age at diagnosis (years)	12.2	11.2
	Tanner’s pubertal stage	III	I
	hSDS	−3.54	−3.72
	TH SDS	−1.75	−2.32
	corr hSDS	−1.79	−1.40
	BMI (kg/m^2^)	20.3	21.5
	HV (cm/year)	4.8	4.0
	StimGH (μg/L)	22.7	31.6
	IGF-I SDS basal	−1.98	−2.39
	IGF-I SDS after IGF-GT	0.80	1.23
	IGF-I/IGFBP-3 molar ratio	0.10	0.09
Therapy duration (years)		1.7	3.1
Data from 1st year of therapy	HV (cm/year)	4.9	5.0
	ΔHV (cm/year)	0.1	1.0
	IGF-I SDS	1.57	2.48
	IGF-I/IGFBP-3 molar ratio	0.52	0.49
Data at the attainment of FH	FH SDS	−3.50	−3.34
	ΔhSDS	0.04	0.38

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
