# Peer review of "Significance of Direct Confirmation of Growth Hormone Insensitivity for the Diagnosis of Primary IGF-I Deficiency"

_jcm, 2020, doi:10.3390/jcm9010240_

Round 1

Reviewer 1 Report

The authors are to be commended for carefully analyzing their data on children who are short, had AGA or SGA at birth and low IGF-I but normal GH responses during testing and response to the IGF generation test. Under current European rules, the SGA subjects could be treated with GH, whereas the AGA subjects might be excluded. However, the authors convincingly demonstrate that they respond equally well but somewhat less than classical GH deficiency and the first year response is a quite good predictor of future growth and final adult height. Thus, a trial of GH treatment in short children born AGA and having criteria outlined above is justified.They do point out that this is a retrospective chart study and that the SGA group consist of only 6 subjects.

1.The conclusion at the end of the paper also should be inculded in the abstract.

2. The English expression is problematic and should be resolved-the paper could be markedly shortened without detracting from the key points and outcomes presented.

3.Have the authors considered molecular diagnostics in the 2 non-responders? No answer is provided about this lack of response, so in contrast to the other subjects.

Reviewer 2 Report

In this retrospective study, the authors analyze the high velocity (HV) response to GH treatment in three groups of children with low stature divided into three subgroups according to 1) small size at birth for their gestational age and IGF-I deficiency (IGFD); 2) normal size at birth and IGFD; 3) GH-deficiency (GHD). From their analysis, they conclude that children with a severe deficit of height, normal GH peak after provocative tests and severe IGFD but normal IGF-I response to GH generation test, may benefit from GH treatment independently of their small or normal size for their gestational age at birth. In their conclusions, they speculate about the possibility of the existence of a non-primary IGFD that should be included in the defects of the GH-IGF-I axis leading to short stature.

In my opinion, since it is a retrospective analysis from the database containing more than 1000 patients treated with GH during the last 20 years, this may explain the several mistakes existing in the analysis carried out, which lead to a difficult interpretation of the results obtained and their conclusions. 

Main problems: 

1) why these children began GH treatment so late in their life? (12-13 years, mean)?. 

2) how was their Tanner's pubertal stage when they began the treatment? this should be included in Tables 1 and 3. 

3) what was their BMI when they began the treatment with GH? this is very important because although GH is the main inducer of liver IGF-I production, the nutritional status plays a key role in the hepatic expression of IGF-I. In fact, if 2-deoxyglucose (which the liver does not metabolize) is administered to liver cell cultures instead of glucose, GH administration is unable to induce IGF-I expression. Moreover, it has been published years ago that anorexia nervosa patients have high pulsatile GH secretion and low IGF-I plasma values. Conversely, obese children lack pituitary GH secretion or it is very low, while plasma IGF-I values are elevated; moreover, these children show a normal or increased growth velocity for their age.

4) how can be explained the effectiveness of GH therapy (line 182) in IGFD patients, leading to all but 2 to the normalization of their final height after 4 years of therapy? and how to explain the increase in IGF-I observed in the GH generation tests when they had been considered as IGFD?.

5) according to these last questions, the manuscript should explain how plasma IGF-I had been measured.

6) the authors should describe, even briefly, what is the role of GH in the fetal size, because it is known that IGF-I and insulin play a major role in fetal growth, and both hormones are stimulated by the glucose that crosses the placenta induced by the maternal IGF-I, which, in turn, is stimulated by the placental GHV. Although the authors do not analyze this concept, they can deduce it from their affirmations in Line 69 and the following lines. 

7) most of circulating IGF-I is bound to IGFBP3, and only about 10% of the total plasma IGF-I circulates free. This free IGF-I is the fraction metabolically active, therefore, the manuscript should describe changes in the IGF-I/IGFBP3 ratio, if they exist, before and after GH treatment.  

The manuscript needs a deep change in its presentation because in its current form it is difficult to be read and lacks scientific discussion in most of its sections.

In Table 2 HV should be indicated as cm/year. 

English has to be corrected, there are many mistakes and misspellings.

The graphical abstract needs to be better explained. 

Round 2

Reviewer 1 Report

The revised version is improved,  but would still benefit from corrections to English expression-I have made minor suggestions for the Abstract which I would like to include

Author Response

Appropriate changes in Abstract are included as suggested. Thank you for your valuable suugestions. 

Kind regards,

Joanna Smyczyńska

Reviewer 2 Report

This manuscript has been improved. Although this is not too important for the study carried out, I would like to know whether after GH treatment Tanner's stage changed. In addition, it would be interesting to know why Tanner's stage was so low for most of the patients in spite of their age. 

The graphical abstract is now more clear to be understood. 

English has some mistakes that should be corrected. 

Author Response

Patients' Tanner stage before treatment was appropriate for their height (height age) and delayed bone age rather than fro chronological age. Nevertheless, in all children puberty started spontaneously and at the attainment of final height they reached at least IV stage of puberty. Appropriate short comment is included in Section 3.2.

Thank you once more for all valuable suggestions and comments.

Kind regards,

Joanna Smyczyńska